# Fine-Tuning Phylogenetic Alignment and Haplogrouping of mtDNA Sequences

**DOI:** 10.3390/ijms22115747

**Published:** 2021-05-27

**Authors:** Arne Dür, Nicole Huber, Walther Parson

**Affiliations:** 1Institute of Mathematics, University of Innsbruck, 6020 Innsbruck, Austria; arne.duer@uibk.ac.at; 2Institute of Legal Medicine, Innsbruck Medical University, 6020 Innsbruck, Austria; Nicole.huber@live.at; 3Forensic Science Program, The Pennsylvania State University, University Park, PA 16802, USA

**Keywords:** mitochondrial DNA, phylogeny, haplogroup, alignment

## Abstract

In this paper, we present a new algorithm for alignment and haplogroup estimation of mitochondrial DNA (mtDNA) sequences. Based on 26,011 vetted full mitogenome sequences, we refined the 5435 original haplogroup motifs of Phylotree Build 17 without changing the haplogroup nomenclature. We adapted 430 motifs (about 8%) and added 966 motifs for yet undetermined subclades. In summary, this led to an 18% increase of haplogroup defining motifs for full mitogenomes and a 30% increase for the mtDNA control region that is of interest for a variety of scientific disciplines, such as medical, population and forensic genetics. The new algorithm is implemented in the EMPOP mtDNA database and is freely accessible.

## 1. Introduction

Estimating the (mitochondrial) haplogroup of a given lineage, also known as ‘haplogrouping’ [1], is crucial to scientific research and diagnostic analyses. Some mitochondrial diseases are observed more frequently on particular haplogroup backgrounds (e.g., [2]), which can be used as screening methods in medical genetics. Reconstructing human migrations has been based on the distribution of extant mitochondrial lineages (e.g., [3]). In forensic genetics, the haplogroup estimate provides information that can be used for phylogenetic and phylogeographic considerations. The phylogenetic background of a lineage serves as plausibility check for the authenticity of the sequencing result, which is particularly important for severely degraded samples that potentially suffer from contamination. The phylogeographic origin of a given sequence can provide investigative leads for law enforcement purposes and assists the statistical interpretation of a database query result by indicating the geographic background of the haplotype [4]. Mitochondrial haplogroup names were developed in the 1990s [5], and a systematically organized catalogue was established in 2009 by Phylotree(mt) ([6]; http://phylotree.org, accessed on 5 May 2021). This freely accessible resource was curated until February 2016, when its last build (17) was released. This version used 24,275 mitochondrial DNA (mtDNA) sequences (=mitogenomes), resulting in a tree with 5435 different haplogroup defining motifs. This tree has been used as a basis for third party software developments that estimate the haplogroup status of a given mitotype (e.g., SAM [7], Haplogrep [8], mtDNAoffice [9], Mitotool [10], EMMA [4], Haplofind [11], Haplogrep2 [12], and SAM2 [13]). With the analysis of new mtDNA population data, an increasing number of sequences is observed that do not truly fit into this tree. Additionally, numerous existing variants present in the current tree have not been used to characterize the respective clades. Therefore, previous work has been performed to enhance the haplogroup estimation by extending the data basis of full mitogenome sequences, e.g., by adding 14,990 GenBank sequences [4]. In this study, we further refine this approach by compiling a database of 26,011 vetted full mitogenomes and by using this database to refine the Phylotree haplogroup motifs. Unambiguous haplogrouping is performed by an alignment-immune algorithm that is independent from alignment and notation of mtDNA sequences and thus provides a strong basis for harmonizing mtDNA nomenclature within and across scientific fields [13].

## 2. Results and Discussion

The revision of the PT17 mitogenomes resulted in 6401 (5435 + 966) revised (rPT17) motifs that can be used for alignment and haplogrouping. This corresponds to an increase of about 18% compared to the original PT17. For the CR, which is commonly targeted in medical, population, and forensic genetics, the increase from 2651 to 3442 haplotypes amounts to 30%.

### 2.1. Haplogroups of the Representative Mitogenomes of PT17

For the evaluation of the new algorithm, all PT17 mitogenome alignments resulted in the nearest estimated haplogroup matching the PT17 motifs or their revisions. The nearest haplogroup equaled the haplogroup in the tree except for the following six cases. For the full mitogenome JQ703290 attached to haplogroup H2a2+(16235), the nearest haplogroups listed were H2a2 and H2a2+(16235), because the mitogenome harbored the ambiguous symbol R at position 16235. Similarly, for the full mitogenome JQ705258 attached to haplogroup H27+16093, the nearest haplogroups were H27 and H27+16093, because the mitogenome harbored the ambiguous symbol Y at position 16093. For the mitogenome DQ112896 attached to haplogroup Q3a+61+62 the nearest haplogroups were Q3a and Q3a+61+62, because of a reduced reading range (436–16021). For the full mitogenome FJ770956 attached to haplogroup M35 the nearest haplogroups were M29′Q and M35, because the private mutations differed only by the transitions 13500C (diagnostic for M29′Q) or 12561A (diagnostic for M35), respectively, which have the same weight of 1.312 due to the ranking in Appendix A of [14]. For the full mitogenome GU122998 attached to haplogroup T2b8 the nearest haplogroup was T2b35, because the private mutations differed only by the transitions 9843G (diagnostic for T2b35) or 3338C (diagnostic for T2b8) with weights of 1.593 and 1.360, respectively [14]. For the full mitogenome AF381985 attached to haplogroup T2e1a1b the nearest haplogroup was T2e1a, because the private mutations differed only by the transitions 15499C (diagnostic for T2e1a) or 16189C (diagnostic for T2e1a1b) with weights of 1.593 and 0.475, respectively [14]. We conclude that the estimated haplogroups were correct and that the mitogenomes DQ112896, GU122998 and AF381985 attached to PT17 should be moved to haplogroups Q3a, T2b35 and T2e1a, respectively.

### 2.2. Approximation Quality of Haplogroup Motifs

Private mutations indicate to what extent a haplotype differs from the motif of its haplogroup. We call a private mutation essential if it is neither present as point heteroplasmy, such as 16093Y, nor a length variant, such as 309.1C. Table 1 shows the average number of essential private mutations relative to the PT17 motifs and their revisions. The maximum decrease (−12%) is found for the representative PT17 mitogenomes, 98% of which are contained in the database of vetted full mitogenomes.

### 2.3. Control Region Sequences

The CR (16024–576) is relatively short, but highly diverse, and has been routinely targeted in medical, population and forensic genetic applications. Table 2 shows the average number of essential private mutations for the haplotypes restricted to CR.

Haplogrouping of CR sequences suffers from the absence of diagnostic coding region mutations, which in general are more reliable to predict a haplogroup status than the diagnostic CR mutations.

Thus, the true haplogroup of a CR sequence corresponds in most cases to a subhaplogroup of the estimated haplogroup. Therefore, we checked whether the haplogroup of the full mitogenome was a subhaplogroup of the estimated MRCA ranks 1 or 2 haplogroups of the CR sequence. This criterion aims at the consistency of haplotype estimates for the CR, but has the drawback that the 966 additional rPT17 motifs for subclades may lead to sharper MRCA rank 1 estimates that may not include the true haplogroup. Therefore, we advise to take MRCA rank 2 estimates into consideration as well in order to arrive at conservative and reliable haplogroup assignments. Table 3 shows the results for our five datasets.

### 2.4. Examples from the Literature

#### 2.4.1. Haplogrouping Examples

To show the effect of the revision of the haplogroup motifs in detail, three complete mitogenomes from GenBank were restricted to the CR and their haplogroups were estimated using only CR mutations. Table 4 summarizes the results. For the mitogenome AP012346.1 from haplogroup F1f, the rPT17 CR estimates were haplogroups F1a and F1f ex aequo, because of the widened motif for haplogroup F1f with 16172Y. For the mitogenome JF824974.2 from haplogroup F1f, the rPT17 CR estimate was F1f because of the subclade motif F1f* bearing both transitions 16172C and 16295T. For the mitogenome JQ705339.1 from haplogroup H4a1a1a, the rPT17 CR estimates were haplogroups H4a1a1a and P+16176 ex aequo, because of the widened motif for haplogroup H4a1a1a with 16176Y.

#### 2.4.2. Alignment Examples

In the first example, the CR sequence USA.154.000067 (Navajo) from [15] is written in the last column of Table 1 as 16111T 16189C 16192.1T 16223T 16290T 16319A 16362C 73G 146C 153G 235G 263G 309.1C 315.1C. The algorithm finds the nearest haplogroup A2+(64)+16189 and rewrites the motif 16111T 16182M 16183M 16189C 16192Y 16223T 16290T 16319A 16362C 16519Y 64Y 73G 146C 153G 235G 263G 315.1c by the transcript M16182A M16183A 16191insC Y16192T Y16519T Y64C 309insC c315.1C to the alignment 16111T 16189C 16191.1C 16192T 16223T 16290T 16319A 16362C 73G 146C 153G 235G 263G 309.1C 315.1C as recommended in [15].

In the second example, the CR sequence USA.ASN.000451 from [15] is written in the last column of Table 1 as 16182C 16183C 16189C 16217C 16247G 16261T 16519C 73G 146C 263G 308T 310d 523d 524d. The algorithm finds the nearest haplogroup B4a1a1 and rewrites the motif 16182M 16183M 16189C 16217C 16247R 16261T 16519Y 73G 146C 263G 315.1c by the transcript M16182C M16183C R16247G Y16519C 308-309delCC c315.1C 523-524delAC to the alignment 16182C 16183C 16189C 16217C 16247G 16261T 16519C 73G 146C 263G 308del 309del 315.1C 523del 524del as recommended in [15].

In the third example, the complete mitogenome JN581642.1 from GenBank has the most parsimonious alignment 456T in the range 451–464 with respect to the rCRS. The algorithm estimates the haplogroup of JN581642.1 as I1a1 and modifies the motif 455.1T of I1a1 by the transcript 459delC to obtain the different alignment 455.1T 459delC, which is less parsimonious, but phylogenetically plausible.

## 3. Materials and Methods

### 3.1. Database of Vetted Full Mitogenomes

To check haplogroup motifs on real data, full mitogenomes were collected from Phylotree (*N* = 24,275 with 17 additional updates from GenBank listed in Appendix A) and extended by quality-controlled mitogenomes of the mtDNA database EMPOP (https://empop.online, accessed on 5 May 2021; [16]; *N* = 3432). All mitogenome sequences with a length shorter than 16,539 bp (30 bp less than the standard length of 16,569 bp) or harboring more than nine ambiguous symbols (e.g., N, Y, M) were excluded. Previous editing of Phylotree mitogenomes [17] were corrected and problematic sequences listed in [17] were not used. That also includes mitogenomes KC993967.1, HM436818.1, HM436819.1 and Young__BJ109 that represent an implausibly large number of yet unreported variants. Thus, the database of vetted full mitogenomes included 22,603 mitogenomes from Phylotree and 3408 mitogenomes from EMPOP, which add up to a total of 26,011 mitogenomes with 22,801 different haplotypes.

### 3.2. Revising Haplogroup Motifs

The original haplogroup motifs included in Phylotree, build 17 (short PT17 motifs) from 18 February 2016 were revised in five consecutive steps. A complete list of the revised PT17 motifs is given in Appendix A.

First, for each haplogroup of PT17, the diagnostic mutations depicted on the tree were translated into EMPOP format and presented as differences to the rCRS. For optional mutations denoted by parentheses on Phylotree branches, ambiguous symbols from the IUPAC code were used, e.g., 16092Y for the optional transition 16092C in haplogroup U2e2. As Phylotree disregards C-insertions at 309 or 315, AC-indels at 515–522, A/C-transversions at 16182 or 16183, C-insertions at 16193 and the C/T-transition at 16519, the PT17 motifs were completed to represent the common full haplotype of the reviewed mitogenomes in all haplogroups. All PT17 motifs were supplemented with 16519Y to ignore the frequent transition T16519C that is not captured in PT17, except for haplogroup B2m bearing 16519A. For all PT17 motifs, the possible insertion 315.1c (c denoting C or gap) was added because the insertion 315.1C is highly abundant, e.g., it is present in almost all EMPOP profiles without long C-stretch (99.53%). All PT17 motifs were extended by the notation 16182M 16183M except for haplogroup B4a1a1a16 bearing 16182T, where only 18183M was added. For PT17 motifs with ambiguous insertions denoted by .XC in Phylotree the dominant type was chosen, as suggested in the revised mtDNA interpretation guidelines published by the International Society for Forensic Genetics (ISFG; [18]).

Second, 23 PT17 motifs were realigned following the rules described in [13,15]. Table 5 shows the affected haplogroups, the number of motifs, if subhaplogroups were included, and the realignments.

Third, 24 PT17 motifs were modified to comply with the reviewed full mitogenomes. Table 6 lists the concerned haplogroups, the edits and the relative frequencies of the mutations in the different haplotypes.

Fourth, a total of 383 PT17 motifs were widened to cover frequent mutations in the reviewed full mitogenomes or in the EMPOP mitogenomes, e.g., the motif of haplogroup W1b1 was extended by 12613R to cover the frequent transition G12613A. For haplogroup A2 and subhaplogroups the motifs were extended by 16192Y to account for the frequent transition C16192T. For haplogroup H4a1a1a and subhaplogroups, the motifs were extended by 16176Y covering the familiar transition C16176T to cope with haplogroup P+16176 in the Control Region (CR). For haplogroup L3h1a1, the motif was extended by 16223Y covering the familiar transition C16223T to cope with haplogroup P5 in the CR. For haplogroup R6 and subhaplogroups, the motifs were extended by 16311Y covering the familiar transition T16311C to cope with haplogroup P6 on the CR.

Fifth, a total of 966 motifs were added to include apparent, yet unnamed, subclades of haplogroups. Following the convention of [19], the subclade motifs are indicated by * and may include subclades of their own. Appendix A lists the mutations characterizing the subclades. For instance, the subclade motif HV* differs from the haplogroup motif HV by the additional transitions 16248T 146C 5460A, and the subclade motif I* differs from the haplogroup motif I by the additional transitions 16311C 1900G 4772C. All subclade motifs differ from the haplogroup motifs by at least three mutations and are represented by at least two different vetted mitogenomes (including sub-subclades, e.g., L2a5*1a in L2a5*1) with the two exceptions C1c+195* and L1b1a16*, which are supported by only one vetted mitogenome. A list of representative mitogenomes for each subclade motif can be found in Appendix A. For the subclade motif L1b1a16* with the additional 9 bp-insertion at 8289, the listed full mitogenome JN214447.1 exhibits length variations 8285insC 8289.5delC inside the 9 bp-repeat and is interpreted as derived from a standard 9 bp insertion. The motif C1c+195* with additional transitions 16176T 203A 204C is supported by the two CR sequences of mitogenomes BRA_18_00000115 and BRA_18_00000168 from the EMPOP database, each bearing all three transitions.

### 3.3. Haplogroup Estimation and Phylogenetic Alignment

For a database query sequence with a given reading frame the proposed algorithm proceeds as follows: First, the algorithm condenses the revised PT17 (rPT17) motifs to the specified range and then searches in the database of condensed motifs for neighbors using the algorithm described in [13] that evaluates costs of private mutations. For instance, for the specified range of the CR, i.e., 16024–576, the 6401 complete motifs are condensed to 3442 CR motifs, which almost halves the size of the database used in the search. Clustering costs by a margin of 0.5 the neighbors of rank 1 and 2 are determined, and the most common recent ancestors (MCRA) of the haplogroups of rank 1 or 2 are computed to provide two single-haplogroup estimates. Finally, the alignment of the nearest neighboring motif is modified by the best transcript to derive an alignment of the query string with respect to the rCRS. The resulting alignment is called phylogenetic, because it resembles the nearest haplogroup motif.

In detail, the alignment algorithm proceeds as follows: In the first step, the transcript is converted into a list of single-symbol substitutions or insertion/deletions (indels) ordered 3′. In the second step, the substitutions and deletions in the list are applied to the alignment and removed from the list. In the third step, the insertions of the current alignment are inserted into the current list in 3′-order and removed from the alignment. In the fourth step, the insertions in the current list are moved 5′ over possible gaps in the current alignment caused by deletions. In the fifth and last step, the insertions in the current list are applied to the current alignment as follows: First, the insertions in the current list on equal positions are combined. If the combined insertions have a joint position not followed 3′ by a deletion, the insertions become insertions in the final alignment. Otherwise, the subsequent deletions are combined, the inserted string is aligned to the deleted string, and this auxiliary alignment decides whether the combined insertions either cancel deletions, change deletions to substitutions or become insertions in the final alignment.

The following theoretical example shows all steps of the alignment algorithm by considering the hypothetical motif 42.1A 64del 65del 73G 309.1C and the query sequence given in a 5′ alignment by 42A 65C 303del. Here, the best transcript is 42delT 63insC 63insC G73A 309–309.1delCC. Step 1 rewrites the transcript to the list 42delT 63insC 63insC G73A 309delC 309.1delC. Step 2 changes the alignment of the motif to 42del 42.1A 64del 65del 309del and reduces the list to 63insC 63insC. Step 3 extends the list to 42insA 63insC 63insC and reduces the alignment to 42del 64del 65del 309del. Step 4 changes the transcript to 41insA 63insC 63insC. Step 5 computes two auxiliary alignments: The inserted symbol A after position 41 is aligned to the deleted symbol T at position 42 by substitution, and the inserted block CC after position 63 is aligned to the deleted block CT at positions 64–65 by match and substitution. Applying these auxiliary alignments to rewrite the deletions yields the final alignment 42A 65C 309del, which is the 3′ version of the alignment used to specify the query sequence.

### 3.4. Evaluation of the Algorithm

To evaluate the algorithm, five data sets were used: the 8268 representative mitogenomes of PT17 that can be retrieved from GenBank via the respective accession numbers; four full mitogenome datasets that were scrutinized in detail by the EMPOP quality control pipeline [16] and belong to the following phylogenetic backgrounds: West Eurasian (*N* = 1179), East Asian (*N* = 2152), Native American (*N* = 548) and Oceanian (*N* = 9).

## 4. Conclusions and Relevance

The reanalysis of the mitogenome sequences that define the current haplogroup nomenclature system of Phylotree(mt) (PT17) led to an increase of discernable subhaplogroups of 18% and 30% for the mitogenome and the CR, respectively. Although the haplogroup nomenclature was not changed, but extended by using the * symbol, its effect on improved alignment and haplogrouping is significant. We note that we deliberately decided to stick to the established haplogroup nomenclature system to avoid confusion in future studies. Alignment to the rCRS by modifying the nearest haplogroup motif does in general not yield the most parsimonious alignment, but reveals the correct private mutations of the query sequence. The proposed revision of the Phylotree motifs and the new alignment algorithm improve the haplogroup estimates and reduce the number of private mutations. The modifications presented in this study have been included in the alignment and haplogrouping software SAM2 [13] that is implemented in the EMPOP database (https://empop.online, accessed on 5 May 2021). EMPOP users thus benefit from more accurate alignment and haplogrouping features that support the harmonization of mtDNA nomenclature and haplogroup estimation across scientific fields.

## Figures and Tables

**Table 1 ijms-22-05747-t001:** Average number of essential private mutations for full mitogenomes.

Data Set	PT17 Average	rPT17 Average
PT17 Representatives	2.51	2.22 (−12%)
West Eurasian	2.29	2.20 (−4%)
East Asian	4.95	4.64 (−6%)
Native American	5.77	5.54 (−4%)
Oceanian	6.44	6.11 (−5%)

**Table 2 ijms-22-05747-t002:** Average number of essential private mutations for CR sequences.

Data Set	PT17 Average	rPT17 Average
PT17 Representatives	0.76	0.65 (−14%)
West Eurasian	0.78	0.73 (−6%)
East Asian	1.70	1.54 (−9%)
Native American	1.85	1.75 (−5%)
Oceanian	3.00	2.89 (−4%)

**Table 3 ijms-22-05747-t003:** Coverage of true haplogroup by CR estimates.

Data Set	PT17 Rank 1	PT17 Rank 2	rPT17 Rank 1	rPT17 Rank 2
PT17 Representatives	8059/8248 (98%)	185/189 (98%)	8092/8248 (98%)	153/156 (98%)
West Eurasian	1115/1179 (95%)	64/64 (100%)	1118/1179 (95%)	61/61 (100%)
East Asian	1956/2152 (91%)	174/196 (89%)	2030/2152 (94%)	104/122 (85%)
Native American	502/548 (92%)	38/46 (83%)	502/548 (92%)	46/46 (100%)
Oceanian	8/9 (89%)	1/1 (100%)	9/9 (100%)	0/0

**Table 4 ijms-22-05747-t004:** Differences of haplogroup estimates in the CR.

Mitogenome	Haplogroup	PT17 Estimate for CR	rPT17 Estimate for CR
AP012346.1	F1f	F1a	F1a, F1f
JF824974.2	F1f	F1a	F1f
JQ705339.1	H4a1a1a	P+16176	H4a1a1a, P+16176

**Table 5 ijms-22-05747-t005:** Realigned motifs.

Haplogroup	PT17 Motif	Realigned Motif
N9b4	16187del 16188T 16189C	16187T 16189C 16193del
J2a1a1a (5)	310.1T 315.1c	311T 315.1c 315.2c
R8b2	456T 456.1T	455.1T 456T
A5b (6)	961C 965.1C	960.1C 961C
M7a2a2		
U4a1a1		
L1c2b1c	5894del 5899.1C	5894C
R11 (6)	8277C 8278.1C	8276.1C 8277C
S3		

**Table 6 ijms-22-05747-t006:** Modified motifs.

Haplogroup	Modification	Frequency	
L5b2	Removed 7972G	A7972G	0/5
L2a1f2	Removed 14566G	A14566G	0/6
L2b1b	Added 183G	A183G	7/7
L3d3a (4)	Changed 16189Y to 16189C, 16278Y to 16278T, 16304Y to 16304C and 16311Y to 16311C	T16189C	13/14
		T16189N	1/14
		C16278T	14/14
		T16304C	14/14
		T16311C	14/14
D5b3a (2)	Removed 16189C	T16189C	0/13
W4b	Added 119C	T119C	4/4
V5	Added 93G	A93G	4/4
H1j9	Changed 16189Y to 16189C	T16189C	2/2
H5c2	Added 16216G	A16216G	3/3
H8c1	Added 9052G	A9052G	3/3
T2a1a1	Changed 143R to 143A and added 8994A	G143A	3/3
		G8994A	3/3
B4b1a2c	Added 310C	T310C	5/5
B4b1a2d	Added 310C	T310C	4/4
B4b1a2f	Removed 16189C and 8281–8289del	T16189C	0/4
		8281–8289del	0/4
B4b1a2g (2)	Removed 16189C and 8281–8289del	T16189C	0/4
		8281–8289del	0/4
B5a2a1b	Changed 16266G to 16266A	C16266A	5/5
U7a4a1a	Changed 16318T to 16318C	A16318C	3/3
K1b1a1b	Changed 16189Y to 16189C	T16189C	1/1
K2a9	Removed 152C and 709A	T152C	0/4
		G709A	0/4

## Data Availability

The data presented in this study are available in the article and in the Appendix A.

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
