# Peer review of "Fine-Tuning Phylogenetic Alignment and Haplogrouping of mtDNA Sequences"

_ijms, 2021, doi:10.3390/ijms22115747_

Round 1
Reviewer 1 Report
-
Author Response
no particular changes required
Reviewer 2 Report
I think that the paper should be published.
I would just appreciate if in the introduction more details about concrete applications of mtDNA haplogrouping would be provided.
Author Response
I think that the paper should be published.
I would just appreciate if in the introduction more details about concrete applications of mtDNA haplogrouping would be provided.
Answer: We added concrete applications in the introduction of mtDNA haplogrouping.Reviewer 3 Report
It is important in research of mtDNA to determine variants on mitochondrial genome correctly and define haplotypes. Researchers have been studying mtDNA by continuously collecting data and applying novel strategies to improve the system of phylogenetic alignment and haplogrouping of mtDNA. I believe that this study is part of that as well.
Overall, it is well organized and described in detail for the process of analysis in this paper, and it is meaningful to look at results after applying new criteria by comparing PT17 and rPT17. I believe this paper can be published without any major revision.
There is a minor question I would like to ask to authors, which is about the result for the East Asian data set in Table 3. The coverage of true haplogroup by CR estimates decreased only in the data set of East Asian after fine tuning (rPT17 rank2). I wonder if more detail explanation can be provided.
Author Response
It is important in research of mtDNA to determine variants on mitochondrial genome correctly and define haplotypes. Researchers have been studying mtDNA by continuously collecting data and applying novel strategies to improve the system of phylogenetic alignment and haplogrouping of mtDNA. I believe that this study is part of that as well.
Overall, it is well organized and described in detail for the process of analysis in this paper, and it is meaningful to look at results after applying new criteria by comparing PT17 and rPT17. I believe this paper can be published without any major revision.
There is a minor question I would like to ask to authors, which is about the result for the East Asian data set in Table 3. The coverage of true haplogroup by CR estimates decreased only in the data set of East Asian after fine tuning (rPT17 rank2). I wonder if more detail explanation can be provided.
Answer:
First, we would like to make a correction:
The entry “184/206” in Table 3 for the East Asian data set is incorrect and has been corrected to “174/196”. As 1956 of the 2152 East Asian CR profiles are covered by the PT17 rank 1 haplogroup estimates, there remain 2152-1956=196 CR profiles, of which 174 are covered by the PT17 rank 2 haplogroup estimates. Thus 1956+174=2130 CR profiles that are covered by the PT17 rank 1 or rank 2 haplogroup estimates, whereas 2030+104=2134 CR profiles are covered by the rPT17 rank 1 or rank 2 haplogroup estimates. The table below summarizes the results for all data sets of Table 3.
The decrease in the East Asian data may be due to the data quality of the sequences, which we are not able to check. However, we can only speculate about this in this review response and do not want to address this topic in the manuscript for understandable reasons.
|
Data set |
PT17 rank 1 or 2 |
rPT17 rank 1 or 2 |
|
PT17 Representatives |
8244/8248 |
8245/8248 |
|
Westeurasian |
1179/1179 |
1179/1179 |
|
East Asian |
2130/2152 |
2134/2152 |
|
Native American |
540/548 |
540/548 |
|
Oceanian |
9/9 |
9/9 |